# The Corey-Seebach Reagent in the 21st Century: A Review

**DOI:** 10.3390/molecules28114367

**Published:** 2023-05-26

**Authors:** Muhammad Haroon, Ameer Fawad Zahoor, Sajjad Ahmad, Asim Mansha, Muhammad Irfan, Aqsa Mushtaq, Rabia Akhtar, Ali Irfan, Katarzyna Kotwica-Mojzych, Mariusz Mojzych

**Affiliations:** 1Medicinal Chemistry Research Lab, Department of Chemistry, Government College University Faisalabad, Faisalabad 38000, Pakistan; haroonfsd144@gmail.com (M.H.); asimmansha@gcuf.edu.pk (A.M.); aqsamushtaq793@gmail.com (A.M.); rabiaakhtar.fsd@superior.edu.pk (R.A.); raialiirfan@gmail.com (A.I.); 2Department of Chemistry, University of Engineering and Technology Lahore, Faisalabad Campus, Faisalabad 38000, Pakistan; sajjad.ahmad@uet.edu.pk; 3Department of Pharmaceutics, Government College University Faisalabad, Faisalabad 38000, Pakistan; manipharma@yahoo.co.uk; 4Department of Chemistry, Superior University, Faisalabad 38000, Pakistan; 5Laboratory of Experimental Cytology, Medical University of Lublin, Radziwiłłowska 11, 20-080 Lublin, Poland; katarzynakotwicamojzych@umlub.pl; 6Department of Chemistry, Siedlce University of Natural Sciences and Humanities, 3-Go Maja 54, 08-110 Siedlce, Poland

**Keywords:** Corey-Seebach reagent, natural products, alkaloids

## Abstract

The Corey-Seebach reagent plays an important role in organic synthesis because of its broad synthetic applications. The Corey-Seebach reagent is formed by the reaction of an aldehyde or a ketone with 1,3-propane-dithiol under acidic conditions, followed by deprotonation with *n*-butyllithium. A large variety of natural products (alkaloids, terpenoids, and polyketides) can be accessed successfully by utilizing this reagent. This review article focuses on the recent contributions (post-2006) of the Corey-Seebach reagent towards the total synthesis of natural products such as alkaloids (lycoplanine A, diterpenoid alkaloids, etc.), terpenoids (bisnorditerpene, totarol, etc.), polyketide (ambruticin J, biakamides, etc.), and heterocycles such as rodocaine and substituted pyridines, as well and their applications towards important organic synthesis.

## 1. Introduction

Elias James Corey is an American chemist well-known for his contribution to the development of methodology and theory of organic synthesis, especially retrosynthetic analysis. He was awarded a Nobel Prize in 1990 for the development of retrosynthetic analysis. His research cooperation with other famous organic chemists has resulted in various name reactions, based on his name, in organic chemistry [1]. One of his famous reactions is the Corey-Seebach reaction which was a combined work of Corey and Dieter Seebach (a German chemist). The Corey-Seebach reagent is formed by the reaction of an aldehyde or a ketone with 1,3-propane-dithiol in the presence of acidic conditions (Lewis acid). Corey-Seebach is a nucleophilic moiety and has widespread applications in various organic transformations. This reaction was first published in 1965, which reported the synthesis of dicarbonyl derivative from 1,3-dithiane [2]. This acyl anion intermediate easily provides access to α-hydroxy ketones [3,4,5,6] by reacting with a range of electrophiles, including carbonyl compounds (Figure 1) [7].

In order to regenerate the carbonyl group that was initially masked when dithiane was utilized as an acyl anion equivalent, it must be hydrolyzed at some point during synthesis. Deprotection has frequently been challenging to accomplish, especially for complicated and sensitive compounds, and as a result, numerous processes have been adopted. The use of traditional methods such as metal salts (mercury(II) chloride [8]) for the deprotection of 1,3-dithiane requires toxic reagents that are generally harmful to the environment. However, there are some facile and efficient methods available in the literature, i.e., 2,3-dichloro-5,6-dicyano-1,4-benzoquinone (DDQ) deprotection [9] and the use of iodine catalyst/H_2_O_2_ [10] which are more environment friendly.

In a typical 1,3-dithiane addition process, 1,3-dithiane is combined with an equimolar quantity of a strong base, such as *n*-butyllithium, and the resultant 2-lithio-1,3-dithiane should serve as an appropriate nucleophile. According to a different procedure described by Andersen et al. [11] the 1,3-dithiane equivalent, 2-trimethylsilyl-1,3-dithiane (TMS-dithiane), could be activated by a stoichiometric quantity of tetrabutylammonium fluoride (TBAF), resulting in the matching carbanion. Corey et al. claim that various cesium salt mixtures that include cesium fluoride may be used as heterogeneous desilylating reagents. There are just a few cases when TMS-dithiane has been activated catalytically, and most of these reactions involve the use of fluoride reagents in equimolar amounts [12].

The Corey-Seebach umpolung technique has been extensively utilized to manufacture a wide variety of natural products such as Swinholide A [13] (**1**, a marine natural product, derived from sponge *Theonella swinhoei*, which shows antitumor and antifungal activity), pironetin [14,15,16] (**2**, derived from *Streptomyces* fermentation broths, which exhibits plant growth regulating action) (Figure 2), ciguatoxin 1B [17] (**3**, one of the main toxins responsible for ciguatera fish poisoning (Figure 3), discovered from moray eel *Gymnothorax javanicus*), and maytansine [18,19] (**4**, shows antitumor activity) (Figure 4). Many synthetic [20] compounds, such as photolabile safety benzoin linkers [21] and bis-3,4-dihydroisoquinolium salts [22], have also been attained using the Corey-Seebach reagent. Earlier, Foubelo et al. published a review article in 2003 concerning the use of 1,3-dithianes in the synthesis of natural products [23]. Until now, the Corey-Seebach reagent has found valuable applications in organic synthesis. Our review article focuses on the utilization of Corey-Seebach reagent in the synthesis of noteworthy natural and synthetic organic compounds reported post-2006.

## 2. Literature Review

### 2.1. Alkaloid-Based Natural Products Synthesis

#### 2.1.1. Lycoplanine A Alkaloids

Lycopodium alkaloids are known to play an effective role in the medication of Alzheimer’s disease [24,25,26]. Over 300 lycopodium alkaloids have so far been isolated, and a number of total syntheses of these alkaloids have been published [27,28,29]. In 2017, Zhao and co-workers [30] first isolated lycoplanine A, a lycopodium alkaloid with the γ-lactone ring. According to biological investigations, lycoplanine alkaloid is a strong inhibitor of the calcium channel (C_av_3.1 T-type) with an IC_50_ value of 6.06 μM. In 2021, Gao et al. [31] reported the synthesis of lycoplanine A isomer by utilizing the Corey-Seebach reagent. To achieve this task, the C=C bond was introduced by using 1,4-dithiane **5** and crotonaldehyde (*E/**Z* > 98%) to afford alcohol **6** in an 88% yield, followed by oxidation to provide the product **7** with an 80% yield. Compound **7** was then treated with Nysted reagent for the introduction of the second C=C, followed by the introduction of fragment A to afford compound **8** by using the Mitsunobu reaction. After a few steps, compound **9** was formed, which upon reaction with Crabtree’s catalyst provided a tetrasubstituted C=C bond product **10** with excellent stereo and regioselectivities. After deprotection of the Boc group, compound **10** was immediately exposed to AcOH, initiating a cascade reaction that produced the stereo-specific cyclized product **11** with a 35% yield. The deprotection of the thioketal group was achieved by using PIFA to afford lycoplanine A **12** isomer with an 83% yield (Figure 1).

#### 2.1.2. Diterpenoid Alkaloids

Diterpenoid alkaloids have been the focus of study by scientists all over the globe because of their fascinating bioactivities and complicated structures [32]. These biologically active compounds were extracted from *Delphinium* and *Aconitum* species that belong to the Ranunculaceae family [33]. In 2017, Min Zhu et al. [34] reported the synthesis of hetidine-type C_20_-diterpenoid alkaloids by utilizing the Corey-Seebach reagent as a key step. For this purpose, 2-lithio-1,3-dithiane species **14** were reacted with iodide **13**, followed by the deprotection of methoxymethyl to afford olefinic phenol **15** with an 80% yield. In the next step, compound **15** was treated with PhI(OAc)_2_, followed by the addition of Sml_2_ to obtain compound **16**, which could then be transformed into the desired hetidine-type diterpenoid alkaloid **17** after a series of reactions (Figure 2).

### 2.2. Terpenoids-Based Natural Products Synthesis

#### 2.2.1. Bisnorditerpene

Diterpenoids are important natural products that display a wide range of chemical diversity and are useful both in medicine and industry. A large number of known diterpenoid compounds are isolated from plants and fungi, and investigations into these species have provided an understanding of their production [35]. In 2010, Pessoa et al. [36] first isolated a bisnorditerpene from *Croton regelianus* var. *matosii*. This herb is utilized in traditional medicine in the Northeastern state of “Caatinga”. In 2016, Xu et al. [37] designed a new strategy for the synthesis of bisnorditerpene by utilizing the Corey-Seebach reagent. To achieve this, an aldehyde **18** was allowed to react with 1,3-propane-dithiol **19** to furnish dithiane **20**, which upon lithiation with epoxide [38] **21** by using the Corey-Seebach reaction afforded precursor **22** followed by the addition of Lewis acid to obtain tricyclic alcohol **23** with a 55% yield. In the next step, secondary alcohol **24** was obtained by desulfurization of **23** with Raney-Ni, followed by the oxidation of **24** with Dess-Martin periodinane (DMP) to afford ketone **25** with a 95% yield. The final process involved the demethylation of ketone **25** with BBr_3_ along with the addition of Phl(OAc)_2_ in CH_3_CN to generate bisnorditerpene **26** (Figure 3).

#### 2.2.2. Totarol Synthesis

Totarol belongs to diterpenes that are found in the sap of *podocarpus totara*, a New Zealand native conifer [39]. The antimicrobial properties [40,41,42,43] of the secondary metabolites in this sap are well known. The wood of this tree displays resistance against rot. Toothpaste and acne medications are just a couple of consumer goods that can contain totarol as an antibacterial ingredient. In 2010, Kim et al. [44] synthesized totarol by utilizing the Corey-Seebach approach as an important key step. The goal of their research was to synthesize totarol diterpenes as a part of a larger research project to determine the mechanism by which tiny molecules could inactivate FtsZ. In order to achieve this, benzonitrile **27** was treated with *i*-PrMgCl to produce compound **28**, followed by the reduction and thioacetal formation to obtain product **29**. In the next step, alkene **30** was synthesized through lithiation of **29** with fragment B followed by alkylation, respectively. Treatment of compound **30** with AD-mix-β afforded regio-isomeric diol **31** with 90–95% enantiomeric excess, and after a few steps, totarolone **32** was formed with a 33% yield. Totarolone **32** was transformed into the desired totarol **33** through the Wolff-Kishner reduction (Figure 4).

### 2.3. Polyketide Based Natural Products

#### 2.3.1. Ambruticin J Synthesis

A significant class of polyketide-based natural compounds known as ambruticin was initially isolated from the bacterium *Sorangium cellulosum* in 1977. They display high biological advantages such as strong antifungal action [45,46,47,48,49,50]. The mechanistic studies of these compounds suggested that ambruticins target Hik1 kinase [51,52] by interacting with fungal osmoregulation. The influence of ambruticin VS3 on soil myxobacteria has recently been studied, and results showed that they are beneficial for the environment by preventing the emergence of antagonistic myxobacterial species. In 2021, Trentadue et al. [53] reported the total synthesis of ambruticin J by utilizing the Corey-Seebach reagent as a key step. For this purpose, dithiane **34** (synthesized from propargyl alcohol) was reacted with epoxide **35** by using the Corey-Seebach reaction to afford compound **36** with a 70% yield, and after a few steps, vinyl iodide **37** was formed. In the following stage, vinyl iodide **37** reacted with pinacol boronic ester **38** through Suzuki coupling, followed by oxidation using Dess–Martin periodinane (DMP) to afford aldehyde **39**. The aldehyde **39** was further treated with fragment C via Julia-Kocienski olefination to afford *E*-olefin **40**, and after a few steps, the desired ambruticin J **41** was formed (Figure 5).

#### 2.3.2. Biakamides

Biakamides are naturally occurring polyketides with significant biological activity [54,55]. In 2017, Kotoku et al. [56] first isolated biakamides from a marine sponge *Petrosaspongia* sp. The purpose of this research project was to isolate marine-based anti-cancer drugs. To achieve this, marine-based biakamides were isolated, and the total synthesis of these drugs has also been described by using the Corey-Seebach reaction in one of their key steps. The synthesis was initiated by using substituted penta-diol **42**, which was converted into corresponding Weinreb amide **43**, followed by reduction with DIBAL to obtain compound **44**. Aldehyde **44** was treated with 1,3-propane dithiol in the presence of iodine to afford 1,3-dithiane **45**. Compound **45** was then allowed to react with alkyl iodide **46** in the presence of *n*-BuLi by using the Corey-Seebach reaction followed by TBAF addition and subsequent tetrapropylammonium perruthenate (TPAP) oxidation to furnish aldehyde **47**. After a few steps, *N*-methyle-neamide **48** was synthesized from secondary amine **49**, followed by the deprotection of 1,3-dithiane to provide compound **50**. The chloromethylene moiety was introduced in the presence of (chloromethyl)triphenyl-phosphonium chloride with *E*/*Z* 3:2 by using the Wittig reaction, which resulted in compound **51**. In the last step, TFA was used for the deprotection of the amine, followed by a condensation reaction with *E*-3-methoxy-2-butenoic acid **52** to afford (4*R*, 6*S*)-biakamides **53** and **54** (Figure 6). The antiproliferative activity of biakamides **53** and **54** was also examined against PANC-1 cell culture (glucose deficient conditions), which provided an IC_50_ value of 0.5 μM.

### 2.4. Photoinitiators

#### 2.4.1. Bisacyldigermanes

The synthesis of improved photoinitiator molecules for free radical polymerization has been a challenging task. So far, a large number of photoinitiators, such as acyl-phosphine oxides, have been successfully synthesized [57]. Among all types, germanium-based photoinitiators are of great importance due to their non-toxic behavior and excellent bleaching properties [58]. In 2022, Wiesner et al. [59] synthesized bisacyldigermanes **59** by utilizing the Corey-Seebach reaction. The purpose of this synthesis was to introduce double germanium content in order to achieve a higher polymerization rate. For this purpose, 1,2-dichloro-1,1,2,2-tetraethyldi-germane **56** was synthesized over four steps from diethyl dichloro germane **55,** followed by lithiation with thioketals **57a**–**e** to afford germane derivatives **58a**–**e**. In the last step, compounds **58a**–**e** were deprotected and oxidized using boron trifluoride etherate and (diacetoxyiodo)benzene (PIDA) to obtain bisacyldigermanes **59a**–**e** in good yields (Figure 7).

#### 2.4.2. Benzoylgermanium Derivatives

Germanium-based photoinitiators have attained great importance due to their high radical polymerization capacity [60,61]. In 2009, Moszner et al. [62] synthesized benzoyl germanium derivatives using the Corey-Seebach reaction. These benzoyl germanium derivatives are used in dental cements and composites. In the first step, aromatic 1,3-dithianes **59** were reacted with *n*-BuLi by using the Corey-Seebach reaction, followed by the reaction with dichlorogermanium compound **60** to afford compound **61a**–**f**. In the last step, compound **61** was dithioketolized in the presence of BF_3_·OEt_2_ and Phl(OAc)_2_ or in the presence of excess iodine and CaCO_3_ in THF to provide PIs **62a**–**f** (Figure 8).

#### 2.4.3. Photoinduced Sensitization

The dithiane-based adducts have been found to be suitable candidates for photoinduced fragmentation [63]. The cleavage of dithiane has been studied by physical approaches such as kinetic isotopic effect [64], Hammett substituent effect, and laser flash photolysis studies [65]. In 2006, Gustafson et al. [66] synthesized benzophenone adducts by utilizing the Corey-Seebach reaction. They also studied computational mechanisms for photoinduced cleavage of dithiane-based benzophenone. For this purpose, the dithianes **63a**–**e** were reacted with benzophenone **64** through the Corey-Seebach reaction to furnish dithiane-based benzophenone adducts **65a**–**e**, followed by photoinduced fragmentation in a temperature range of −40 °C–40 °C (Figure 9).

#### 2.4.4. Photoinduced Bis-Addition

The Corey-Seebach methodology, which is built on lithiodithiane reactions with various electrophiles, particularly carbonyl compounds, has taken a leading position among many of the traditional modern synthetic chemistry strategies [67]. One of its variants, the methyl dithiane addition with benzoyl chloride or alkyl benzoates, provides access to tertiary alcohols with two dithiane moieties [68]. Valiulin et al. [69] reported a synthesis of dithiane adducts by using the Corey-Seebach reaction. The methodology involved the reaction of alkyl dithiane methyl benzoate or benzoyl chloride to afford the target molecule. It was observed that the dithiane adduct **67** was only formed when methyl-containing benzoyl dithiane **66** was used. The acetophenone tethered thio-ortho ester **68** was formed when the R group with the higher substitution was used (Figure 10).

### 2.5. Bisbibenzyl Analogue

#### Riccardin C Synthesis

Bisbibenzyls are important natural products that are found in the bryophytes, such as liverworts [70,71,72]. Among these, riccardin C has gained great importance due to its effectiveness against cardiovascular diseases. Riccardin C also shows antifungal [73], anti-bacterial, and cytotoxic activity [74]. In 2016, Almalki et al. [75] purposed the total synthesis of riccardin C by using the Corey-Seebach macrocyclization strategy. For this purpose, compound **70** was synthesized from catechol **69** over a few steps, followed by the reaction of compounds **70** and **71** via Suzuki-Miyaura coupling to obtain compound **72**. Further, compounds **73** and **74** were reacted to form biaryl ethers **75**, followed by coupling with aldehydes **75** and alcohol **72** to afford compound **76**. Compound **76** was transformed into **77** by the Heck reaction followed by the reduction of alkene using diimide. Next, SOCl_2_ or MsCl was used to convert the alcohol into chloride **78**. In the last step, dithiane was deprotonated by using *n*-BuLi at 78 °C and the Corey-Seebach reaction to afford macrocycle **79**, followed by the deprotection of benzyl ether and dithiane to achieve riccardin C **80** (Figure 11).

### 2.6. Biocompatible Polyesters

#### Benzoin-Derived Diol Linker

The synthesis of photodegradable biocompatible polymers has created a serious problem due to their slow degradation and unwanted by-products. Such restrictions may be overcome by using dithiane-protected benzoin derivatives [76,77]. In 2018, Englert et al. [78] synthesized diol benzoin derivatives that act as active monomers for the polymerization process. The main focus of this study was to synthesize micro and nanoparticles that may release compounds on demand within predetermined time frames when “opened” by UV radiation. For this purpose, a diol precursor **82** was synthesized from 3-hydroxybenzaldehyde **81** in three steps by using the Corey-Seebach reaction. In the next step, product **82** was activated to obtain compound **83** or by the reaction of butyryl chloride with compound **82** to afford product **84** prior to activation. In the last step, the polymerization of compound **83** was done with adipoyl dichloride to attain polyester **85** through polycondensation (Figure 12).

### 2.7. Tetraphenylcyclopentadienones

Tetraphenylcyclopentadienones are very important diene substrates that have been widely used in different product syntheses such as photochromic benzoyranes quinonoid intermediates and graphene intermediates [79,80]. In 2017, L. Prati et al. [81] synthesized new 1,3-diarylphencyclones by utilizing the well-known Corey-Seebach reaction. The purpose of this research was to present a stereodynamic and conformational study of 1,3-diaryl-phenylclones to obtain stable atropisomers. For this purpose, 1,3-dithiane **86** was allowed to react with benzyl derivatives by using a double Corey-Seebach reaction followed by the deprotection with NaHCO_3_/I_2_ to obtain 1,3-diarylketones **87**. In the next step, compound **88** was reacted with 1,3-diarylketones **87** to afford 1,3-diaryl-phencyclones **89a**–**e** (Figure 13). Among all the synthesized compounds, **89d** and **89e** were found to exhibit exceptionally stable atropisomers (racemization energy > 35 kcal/mol).

### 2.8. Scleropentaside A

Scleropentasides are a prime class of natural products that have been extracted from the twigs, leaves, and stems of *dendrotrophe frutescens* and *scleropyrum pentadrum* [82], respectively. Both plants exhibit a variety of uses in traditional Asian medicine, including skin treatments, rheumatic pain relief, etc. This innovative class of natural compounds has an unrivaled anomeric-*β*-glycosidic motif and a furan ring [83]. One of the most iconic members of this class is scleropentaside A, which displays a radical scavenging activity. Boehlich et al. [84] reported an exclusive and general method for the preparation of acyl-glycosides by utilizing the Corey-Seebach reaction. This methodology has been exclusively used for the short synthesis of scleropentaside A. For this purpose, a silane-protected carbohydrate **90** was reacted with furfural dithiane **91** to obtain product **92**, which upon deprotection, afforded sceropentaside A **93** with a good yield (Figure 14).

### 2.9. Steroids

#### 2.9.1. Withanolide A

One of the most potent components in the methanolic extracts of *ashwagandha* is withanolide A, which was isolated from the roots of *Withania somnifera* [85]. It has been shown to have strong pharmacological properties with regard to neurite regeneration, axonal outgrowth, and repair of damaged synapses in mice [86,87,88,89]. The synthesis of withanolide A has been presented with various synthetic problems. In 2013, Liffert et al. [90] reported a new method for the synthesis of withanolide A by using the Corey-Seebach reaction in one of their key steps. For this purpose, the pregnenolone **94** was protected with TBS, followed by lithiation with dithiane to obtain **95** by using the Corey-Seebach reaction. In the next step, the deprotection of dithiane was achieved by using (*N*-Chlorosuccinimide) NCS in the presence of dichloromethane (DCM) followed by MOM protection of the OH group to afford aldehyde **96**. Aldehyde **96** was treated with vinylogous enolate and LiHMDS in the presence of DMPU and THF, which resulted in the synthesis of unsaturated lactone **97** with an 87% yield and excellent stereoselectivity (dr 93:7). After a few steps, unsaturated enone **98** was formed, which upon treatment with H_2_O_2_ (Triton B), hydrazine, and PDC afforded withanolide A **99** in a 30% yield (Figure 15).

#### 2.9.2. Cholanic Acid Derivatives

Cholanic acid is one of the important intermediates for many reactions. It was extracted from a sea pen by Djerassi et al. [91,92]. In 2007, Shingate et al. [93] reported the stereoselective method for the synthesis of 20-*epi* cholanic acid derivatives from dehydropregnenolone acetate by using the Corey-Seebach reaction in one of their key steps. The synthesis of cholanic acid was initiated from the chemo-selective catalytic hydrogenation of 16-dehydropregnenolone acetate **100**, followed by hydrolysis to obtain compound **101**. After a few steps, compound **102** was formed, which was further reacted with 1,3-dithiane using the Corey-Seebach reaction to afford compound **103** and side product **104** with a 77% and 4% yield, respectively. In the next step, compound **103** was treated with SOCl_2_ and pyridine in the presence of DCM to obtain ketene **105** with an 84% yield, which provided iso-methyl ether **106** after a few steps. In the last step, the deprotection of compound **106** was carried out to afford 20-*epi* cholanic acid derivative **107** (Figure 16).

### 2.10. Rodocaine

Rodocaine is an important chemical that is used in ophthalmic anesthesia [94,95,96]. There are a number of methods that have been developed for the synthesis of this molecule. In 2017, Meyer et al. [97] developed a new method for the enantioselective synthesis of rodocaine through enantioselective hydroazidation by utilizing the Corey-Seebach reaction in one of their key steps. For this purpose, compound **108** was converted into cyclopentene **109** through Malaprade glycol, thioacetalization, and the Corey-Seebach reaction, respectively. Compound **109** was then subjected to enantioselective hydroazidation in the presence of (–)-IpcBH_2_ to afford trans azide **110** with a 61% yield (*er* 75:25), followed by desulfurization and Boc protection to furnish compound **111**. In the last step, compound **111** was alkylated with iodide **112** to afford the impure rodocaine with a 68% yield (er 74:26), which was precipitated with H_2_O/MeOH to obtain pure enantiomeric rodocaine **113** with a 15% yield (Figure 17).

### 2.11. D-Glucosamine Trimethylene Derivatives

The extension of the carbohydrate chain by using the Corey-Seebach method (dithiane chemistry) is well known [98]. A number of D-glucosamine-based trimethylene derivatives have been synthesized, but these approaches have some limitations regarding the tolerance of leaving groups. In 2006, Chen et al. [99] utilized trimethylene acyl-d-glucosamine to synthesize α-imidate dithiane. To achieve this, compound **114** was treated with *n*-BuLi to produce stabilized imidate-carba dianion **115**, followed by the addition of D_2_O to afford compound **116**. The iodination of compound **114** was also performed to check the reactivity, and two different products, **117** and **118**, were obtained in different conditions. The carbon atom extended carbohydrate **119** was formed by the reaction of compound **114** with *n*-BuLi, followed by the reaction with DMF, ethyl chloroformate, and methyl chloroformate. Compound **114** was also treated with cyclohexanone or cyclopentanone to yield compounds **120** and **121**. In addition to this, compound **114** was also reacted with substituted d-ribofuranose-3-ulose **122** to afford compound **123** with an average yield (up-to 28%) (Figure 18).

### 2.12. (−)-Calystegine B_3_

As effective and selective glycosidase inhibitors, carbasugars and azasugars are among the most appealing compounds in the world of *N*-carbohydrates [100]. The majority of the compounds contain five- or six-membered fused-ring structures. Most of them are employed as chemotherapeutic agents to treat viral infections and diabetes. The calystegine compounds belong to the Solanaceae family [101,102], and their analogs have been created since they are thought to be pioneer compounds for novel bioactive drugs. Chen et al. [103] synthesized (−)-calystegine B_3_ from D-glucosamine-based trimethylene dithioacetal by virtue of the Corey-Seebach reaction. Their methodology involved the synthesis of compounds **124** and **125** from D-glucosamine followed by epoxidation to acquire diastereoisomers **126a**, **b** and **127**, respectively. The anionic cyclization of compounds **126a**, **b** was performed using the Corey-Seebach reaction to afford carba-analogs **128a**, **b** (*ca*. 1.5:11, 83%). Similarly, compound **127** cyclized through the Corey-Seebach methodology to obtain carba-analogs **129** and **130** (*ca*. 2.4:1, 78%). In addition, compound **130** was transformed into ketone **131** in a few steps, followed by Pd-C [104] addition in the presence of THF and *O*-Bn deprotection to yield the title compound (−)-calystegine B_3_
**132** (Figure 19).

### 2.13. Substituted Pyridines

The nitrogen-containing aromatic heterocycles are essential components in pharmaceutical and natural products [105]. Substituted pyridines are extremely evident among them. For three decades, the synthesis of pyridine has gained a lot of attention of from the scientific community [106]. Despite the fact that there are numerous chemical ways to generate such heteroaromatics, there is still a great deal of interest in finding new approaches that would provide quick and precise exposure [107]. Chen et al. [108] developed an efficient method for the synthesis of substituted pyridines by utilizing the Corey-Seebach reaction. The methodology involved the reaction of dithiane **133** with α,β-unsaturated ketones or aldehydes **134** to obtain a series of trisubstituted alkenes **135** in good to excellent yields (62% minimum and 96% maximum). These trisubstituted alkenes **135** underwent Ti-mediated coupling with substituted aldehydes **136** to furnish substituted pyridines **137** with average to good yields (33–82%) (Figure 20). The role of alkene geometry in this reaction was also explored, and it was observed that the geometry of alkenes did not play any crucial role in the synthesis of pyridines.

## 3. Conclusions

This review article provides a thorough analysis of the synthesis of natural and synthetic compounds which involve the Corey-Seebach reaction as a major step in their synthetic methodologies. The Corey-Seebach reagent is of particular importance in organic synthesis owing to its ability to generate valuable chemical derivatives from basic, easily accessible starting materials. We, therefore, come to the conclusion that Corey-Seebach reagent serves as a synthetic equivalent as well as a protective group for the carbonyl functionality and has extensively been employed in the synthesis of natural products such as lycoplanine A, bisnorditerpene, totarol, ambruticin J, biakamides, as well as synthetic molecules (Bisacyldigermanes, photoinitiators, benzoin-derived diol linkers, substituted pyridines). With the fact that the Corey-Seebach reaction has been the subject of extensive investigation, we anticipate that this analysis will spur synthetic scientists to develop fresh approaches and innovative theories in this area.

## Data Availability

All data is contained in the manuscript.

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
