# Peer review of "The Corey-Seebach Reagent in the 21st Century: A Review"

_molecules, 2023, doi:10.3390/molecules28114367_

Round 1
Reviewer 1 Report
The review manuscript submitted by Mojzych and co-workers reports an analysis of applications of the Corey-Seebach reagent after 2006 in the synthesis of valuable compounds. The review is interesting and can be of interest for many research groups. However, I have several suggestions and comments as follows, to be addressed before publication. Why does the analysis describe articles only after 2006? It is not clear in the main text if other review articles are reported in literature. I also suggest to report in the introduction a section about the methods for deprotection of the 1,3-dithiane derivatives. This would help the readers.
Structures have to be checked as for example 30 (a double bond missing), 122, 143 (a double bond missing).
It would be nice to see also the photoinduced fragmentation in the scheme 9.
Try to explain as much as possible acronyms as DBH at line 394 and describe in the scheme.
In Scheme 22, show the nitrogen source.

Dear Editor,
the English language seems fine, only a few typos.
Best regards
Author Response
To whom it may concern:
Thank you very much for peer reviewing our manuscript and we appreciate your complimentary recommendations as your comments have helped us significantly to improve the manuscript.
Regards
Mariusz Mojzych

Reviewer 2 Report
The review article “The Corey-Seebach Reagent in 21st Century: A Review” covers the most recent applications of Corey-Seebach reagent in the total synthesis of natural products. Indeed, the Corey-Seebach reaction has been extensive utilized by chemistry community as a critical tool to include a carbonyl group in organic molecules. A review to cover this area is of demanding and of interest not only for chemistry community but also for its further development.
The authors thoroughly investigated, and categorized the related papers. A sound introduction story about each subcategory has been told in the review. In all the cases depicted, the chemcal structure has been well drawn and reorganized with reaction conditions and yields included. In conclusion, it is a comprehensive review with a good amount of work. It is of interest to the chemistry community and the Molecules’ readers. However, there are some concerns that must be addressed. Therefore, this reviewer recommends the reconsideration of its publication on Molecules after major revision.
General comments:
1) The 1,3-dithane deprotection is also considered a crucial step after Corey-Seebach reaction and the reaction conditions used vary in each case. Please include this step in all the cases and the corresponding discussion section in the context.
2) The chemical abbreviations (e.g. DCM, DMP, TPAP, NMO…) used in the context and the scheme should give the full name in the context for the readers to follow.
3) Please double-check the synthesis description. In most of the cases rewrite or revision is needed. Some are shown in specific comments.
Specific comments:
Page 2, Figure 1: Lithium counterion should be included in the scheme with carbonion. The product should be alcoholate with lithium counterion. There should be a reference of figure 1 in the context.
Page 2, Line 58: “Dangerous fluoride” is a very ambiguous term. Please be specific.
Page 4, Line 106: compounds 13 and 14 were labeled incorrectly.
Page 5, Scheme 2: “NaHCO3” was incorrect.
Page 5, lines 123 and 124 need to be rewritten.
Page 9, Scheme 6: the structure of 47 is incorrect. Please revise. Thereby, page 8 line 47 should be rewritten.
Page 10, Line 202: compound 55 was labeled incorrectly. Line 205, diethyl etherate should give the full name of it. Line 219, “phI(OAc)” is incorrect. Line 220: “pls” is incorrect.
Page 11, Scheme 8: “Ph(OAc)2” is incorrect. Line 244, “double lithiation” is a wrong term. It didn’t appear like a double lithiation.
Page 12, Scheme 10: please define ‘X’ and ‘Y’. Line 261, “in the next step” is incorrect. It’s not the following step.
Page 13, Scheme 11: condition b should be deleted. Conditions “2) 77, TsNHNH2, NaOAc, THF, H2O” was labeled in the incorrect reaction.
Page 18, lines 373 and 374 need to be rewritten.
Page 18-19, chapter 2.11: this example needs to be reconsidered whether it is appropriate to be classified as a Corey-Seebach reaction.
Page 19-20, chapter 2.12 and scheme 19: the structure of 121 is incorrect. Please double-check with the original paper (ref 94). The reactions depicted in scheme 19 couldn’t be found in the paper.
Page 21, Scheme 20: the yield of 138 and 139 is very confusing.
Page 22-23, please double check with the original paper. The Corey-Seebach reaction couldn’t be found in the paper.
Author Response

(The authors gave the same response as above.)

Reviewer 3 Report
The author has succeeded in writing a useful review of recent advances in the applications of the Corey-Seebach reactions towards to the total synthesis of natural products.
This paper not only provides various examples of substrate adaptations of the Corey-Seebach reactions, but also provides information on the subsequent deprotection reactions. This manuscript was well written and will be interest to readers in this field. I think this work is acceptable after minor revision. My minor comment shown below.
Comment:
P.11, Line 236, Figure of structural formula about (65a-e).
R group may be added to the structural formula.
P. 23, Line 459. Scheme 21: Synthesis of tetracyclines (141).
(141) may be fixed to (150).
Author Response

(The authors gave the same response as above.)

Reviewer 4 Report
The paper submited for review:
The Corey-Seebach Reagent in 21st Century: A Review.
Authors: Muhammad Haroon , Ameer Fawad Zahoor, , Sajjad Ahmad , Asim Mansha , Muhammad Irfan , Aqsa Mushtaq , Rabia Akhtar, Ali Irfan , Katarzyna Kotwica-Mojzych and Mariusz Mojzych
The Review article submited for review meets the requirements for this type of work.Authors presents the synthesis of wide spectrum of natural products and synthetic molecules with using Corey-Seebach reagent. There are presented preparation methods for the following group of the active compounds like: Alkaloids (lycopodim alkaloids and diterpenoid alkaloids), Terpenoids, Diterpenoids, Diterpenes, Poliketides, Photoinitiators (eq benzoylgermanium derivatives), Bisbibenzylic analoge (eq Riccardin C), Photodegradable biocompatible polyesters (eq benzoin-derived diol linkers), Tetraphenylcyclopentadienones, Scleropentasides, Steroids, Rodocaine, Proline derivates, D-glucosamine derivatives, Solanaceae family compound [eq. (-)-Calystegine B3], Tetracyclines and Substituted pyridines. Of course,such a large amount of material presented in the work (30 pages, 22 schemes, 4 figures, near 500 lines of the text and over 100 cited references) increased the like likelihood of mistake or inaccurances, what just happened. However,this does not change the high evaluation of the work in therms of didactic,cognitive and publishability.
In Abstract the Authors write that the review article is mostly based and focused on post 2006 literature.Unfortunately,this is inacurate,because from 103 cited papers,more than half (53 to be exact) were published before 2006.Please correct and/or comment above.The Authors did not avoid another errors and inaccurace that should be corrected.Here they are:
In the schemes 3 ( Synthesis of bisnorditerpeneand ) and scheme 8 (Synthesis of benzoylgermanium derivatives) the structures of the compounds shown,do not correspond to the sizes of the compounds in the other schemes.They are too small (probably copied) or big and should be shown all the same size from the first to the last scheme.
Line 163 could you comment and explain „after a few steps vinyl iodide (37) was formed”,
Line 167 similar remark „after a few steps, the desired ambruticin J (41) was formed”.The presentation of above details is of great didactic and educational importance especially for the young generation of the chemical community.
In contrast,the total synthesis of (4R,6S) Biakamide A is shown exemplary.
Line 191 in „Biakamides (54)” should be Biakamides (53) and (54).
Line 219 there is „phI(OAc) should be PhI(OAc).
Line 264 instead „used toconvert alcohol ”,shoul be used to convert alcohol.
Author Response

(The authors gave the same response as above.)

Round 2
Reviewer 1 Report
After revisions, the manuscript can be now accepted for publication.
Best regards

Dear Editor,
in my opinion the language is fine and comprehensible.
Best regards
Author Response
-
Reviewer 2 Report
Thanks for the author's careful revision. The raised concerns have been addressed. Therefore, this reviewer recommends its publication on Molecules.
Author Response
-